# Prevalence, Knowledge and Awareness of Pelvic Floor Disorder among Pregnant Women in a Tertiary Centre, Malaysia

**DOI:** 10.3390/ijerph19148314

**Published:** 2022-07-07

**Authors:** Mukhtar Nur Farihan, Beng Kwang Ng, Su Ee Phon, Mohamed Ismail Nor Azlin, Abdul Ghani Nur Azurah, Pei Shan Lim

**Affiliations:** 1Department of Obstetrics and Gynaecology, Hospital Tuanku Fauziah, Jalan Tun Abd Razak, Kangar 01000, Perlis, Malaysia; nurfarihan83@gmail.com; 2Department of Obstetrics and Gynaecology, Faculty of Medicine, Universiti Kebangsaan Malaysia, Jalan Yaacob Latiff, Bandar Tun Razak, Cheras 56000, Kuala Lumpur, Malaysia; sephon88@yahoo.com (S.E.P.); azlinm@ppukm.ukm.edu.my (M.I.N.A.); azurah@ppukm.ukm.edu.my (A.G.N.A.); peishan9900@yahoo.com (P.S.L.)

**Keywords:** delivery, knowledge, pelvic floor disorder, pelvic organ prolapse, urinary incontinence

## Abstract

Pelvic floor disorders are common and of concern, as the majority of maternity healthcare providers seldom discuss this issue with patients compared to other antenatal issues. The aim of this study is to determine the prevalence and to assess the knowledge and awareness of pelvic floor disorder (PFD) among pregnant women in a tertiary centre in Malaysia. We also aim to assess the association between women’s risk factors regarding their knowledge and awareness of pelvic floor disorder so that primary prevention strategies can be planned, initiated and implemented in the future. This is a cross-sectional study with a total of four hundred twenty-four pregnant women that were recruited over a 6-month duration from May to November 2017 in a tertiary centre in Malaysia. The Pelvic Floor Distress Inventory (PFDI-20) was used to evaluate pelvic floor distress symptoms during pregnancy, namely urinary incontinence, pelvic organ prolapse and faecal incontinence. A validated Prolapse and Incontinence Knowledge Questionnaire (PIKQ), which consists of 24 items, was used to assess respondents’ knowledge about urinary incontinence (UI) and pelvic organ prolapse (POP). A total of 450 participants were approached, but 424 participants agreed to participate, showing a response rate of 94.3%. The median age was 31.5 years old, and 33.3% were primiparity. Overall, 46.1% of pregnant women had at least one symptom of pelvic floor disorder during pregnancy. Of these, 62.3% experienced urinary incontinence, 41.1% experienced symptoms of pelvic organ prolapse, and 37.8% experienced symptoms of faecal incontinence. The overall median score of PIKQ was 12.0 (8.0, 17.0). The median score for PIKQ—UI was 7.0 (5.0, 9.0) and the median score for PIKQ—POP was 6.0 (4.0, 8.0). There were 341 (80.4%) pregnant women that had a low level of knowledge in UI, and 191 (45.0%) had a low level of knowledge in POP. Having a tertiary level of education and receiving antenatal specialist care were both associated with better proficiency in both the PIKQ—UI (*p* < 0.001) and PIKQ—POP (*p* < 0.001) subscales. Pelvic floor disorder during pregnancy was common. A knowledge of pelvic floor disorder was lacking among pregnant women in this study. Having a tertiary education and receiving antenatal specialist care were both associated with better knowledge proficiency. This study hopefully serves as a basic platform for future educational programs to bridge the knowledge gaps in PFD among pregnant women.

## 1. Introduction

The pelvic floor is a group of muscles and ligaments that supports the pelvic organs. Pelvic floor disorder (PFD) comprises a group of disorders that often coexist. These include pelvic organ prolapse (POP), urinary incontinence (UI), faecal incontinence (FI), pelvic pain and sexual dysfunction [1]. The prevalence of at least one PFD increases with age, i.e., from 9.7% in women aged between 20 and 39 years to 47.7% in those aged 80 and above [2]. The lifetime risk of a woman undergoing surgery for PFD is 20% by the age of 80 years old [3]. PFD not only causes significant disturbances psychologically and socially, it also significantly impairs the woman’s quality of life [4]. PFD has raised public health concerns in recent years due to increasing health expenditure as well.

The research has focused on preventive measures, as treatment of PFD does not guarantee a long-term cure. PFD are multifactorial, with pregnancy and delivery being the most common risk factors. It has been reported in 15.4–24.2% of women having bothersome lower urinary tract symptoms at 36 weeks gestation, with 10–15% of symptoms persisting one year after delivery [5]. Another multicentre study with 1500 participants showed that the prevalence of UI was at 10.8% twelve months prior to index pregnancy, and the risk increased to 55.9% by the third trimester [6]. Other studies have reported the prevalence of UI during pregnancy ranging from 19–42% [7]. A study performed by Gao L. et al. described the pelvic floor function of the primigravida in the first trimester and confirmed that a larger right-to-left diameter of the levator hiatus and the use of a squatting type of toilet were significantly associated with pelvic floor muscle weakness [8].

Despite mounting evidence that PFD is common and of concern, the majority of maternity healthcare providers rarely focus on this issue during patient care as compared to other antenatal issues such as anaemia or medical disorders [9]. Young women also seldom received formal or informal information about PFD. A study by Parden et al. showed that only 17.7% of their respondents had ever received education about UI; 23.8% were aware of FI, and 21.2% were familiar with POP [10]. However, only 15% of patients suffering from stress urinary incontinence (SUI) sought help, and only one third of them received the recommended treatments [11]. Therefore, providing knowledge and education to both women and healthcare providers would enhance the awareness of PFD and further strengthen prevention strategies. 

This study aimed to evaluate the prevalence of PFD among pregnant women in Malaysia, as the data are still scarce. Second, this study aimed to assess the knowledge and awareness of PFD so that primary prevention strategies can be planned, initiated and implemented in the future. 

## 2. Materials and Methods

### 2.1. Study Design

This was a cross-sectional study, over a 6-month duration, from 25 May to 24 November 2017, at a tertiary centre in Malaysia. It was approved and funded by the Medical Research and Ethics Committee of Universiti Kebangsaan Malaysia Medical Centre (research code: JEP-2017-344). All pregnant women aged above 18 years attending the antenatal clinic or admitted through the patient admission centre were approached. Those that fulfilled the inclusion criteria were recruited. Respondents were explained about the study and informed consent was taken. Demographic data such as age, ethnicity, gravidity, parity, gestational age, previous mode of delivery, previous abdominal or gynaecological surgery, maternal education level, occupation, attendance of prenatal classes and level of antenatal care were obtained.

Respondents were requested to answer two sets of questionnaires, namely the Pelvic Floor Distress Inventory-20 (PFDI-20) and the Prolapse and Incontinence Knowledge Questionnaires (PIKQ). The time taken for answering the questionnaires was around 10–20 min. 

### 2.2. Instruments and Data Collection

Diagnosis and severity of PFD such as UI, FI and POP were assessed using the validated PFDI-20 [12]. The PFDI consists of 46 questions separated into 3 scales, the UDI, Pelvic Organ Prolapse Distress Inventory (POPDI), and the Colorectal-Anal Distress Inventory (CRADI). The PFDI has been shown to have good test–retest reliability (intraclass correlation coefficients (ICCs) of 0.86) and excellent internal consistency (Cronbach’s alpha of 0.88). The scales of the PFDI demonstrate significant association with appropriate measures of symptom severity and pelvic floor diagnoses, thereby demonstrating construct validity [12]. The shortform version of the PFDI has a total of 20 questions and 3 scales (UDI-6, POPDI-6, and CRADI-8). Each of the 3 scales of the PFDI-20 is scored from 0 (least distress) to 100 (greatest distress). The sum of the scores of these 3 scales serves as the overall summary score of the PFDI-20 and ranges from 0 to 300 [12].

Respondents’ knowledge about UI and POP were assessed by a previously validated Prolapse and Incontinence Knowledge Questionnaire (PIKQ), a 24-item questionnaire that includes 12 questions that focus on knowledge on UI (UI scale) and 12 questions focusing on the knowledge of POP (POP scale) [13]. Each question has 3 options: Agree, Disagree and Not Sure. For the questionnaire, true answers are scored as 1 each, while false and ‘Don’t know’ answers are scored as 0. A score of at least 10 out of 12 on the UI subscale and 6 out of 12 on the POP subscale are considered as having proficiency in the knowledge.

### 2.3. Data Analysis

SPSS (Statistical Package for Social Science) version 23 was used for data analysis. Participants’ profiles were presented descriptively in terms of frequency and percentage, by mean and standard deviation (for normally distributed data), or by median and interquartile range (for non-normally distributed data). Maternal characteristics and level of knowledge (PIKQ—UI and PIKQ—POP) analyses were performed using Pearson’s Chi-square test. The significance level was set at *p* < 0.05.

## 3. Results

A total of 450 participants were approached, and finally, 424 participants agreed to participate, showing a response rate of 94.3%. The median age was 31.5 years. The characteristics of the female participants in the study are given in Table 1.

### 3.1. Prevalence of Pelvic Floor Disorders among Pregnant Women

The results of the PFDI-20 questionnaire are presented in Table 2. The most prevalent item group was UDI-6, in which the prevalence within this group was as high as 62.7%. This was followed by POPDI-6 (41.8%) and CRADI-8 (37.8%) (Table 2). When analysing all the items in detail, almost half of the participants (54.7% and 48.3%) described having pressure and heaviness in the lower abdomen and pelvis. Thirty-five percent of women had a bulge that could be seen or felt in the vagina. One third of women experienced incomplete bladder emptying, and 26% of them needed to push up on the bulge in order to complete their urination (Table 3). With regard to bowel symptoms (CRADI-8), around 30% had faecal incontinence and up to 5% described this as bothersome and affecting them. Almost 50% of the participants had bowel urgency, and 20% of them described this as bother them moderately or quite a bit (Table 3). Lastly, 62.7% participants experienced urinary frequency and 49% of them had urge incontinence. Fifty-four percent had urinary stress incontinence, 35.8% had dyspareunia, and 43.9% described a reduction in sexual interest (Table 3).

### 3.2. Knowledge and Awareness of Pelvic Floor Disorders among Pregnant Women

Overall, the median score for knowledge on pelvic floor disorders was 12, which was considered low (cut off point of 16). The median knowledge scores for PIKQ—Urinary Incontinence and PIKQ (Table 4)—Prolapse were 6.0 and 7.0, respectively (Table 5). Three hundred forty-one (80.4%) pregnant women had a low level of knowledge on UI and 191 (45.0%) had a low level of knowledge about pelvic organ prolapse (POP) (Table 6).

Pregnant women with tertiary education were more likely to have a proficiency in the knowledge for PIKQ—UI (*p* = 0.000) and PIKQ—POP (*p* = 0.000) as compared to secondary education and below. Those who received antenatal specialist care also appeared to have proficiency in the knowledge for both PIKQ—UI (*p* = 0.000) and PIKQ—POP (*p* = 0.000) as compared to those who were taken care of by a local clinic or medical officer/trainee. There is no significant difference in the level of knowledge according to age, primiparity, marital status and attending antenatal class [Table 7].

## 4. Discussion

### 4.1. Prevalence of Pelvic Floor Disorders among Pregnant Women

Pelvic floor disorder is a public health concerns, as it affects one’s quality of life with significant psychosocial effects, including low self-esteem, anxiety, frustration and depression [14,15]. Despite the prevalence, economic burden, and psychosocial and physical morbidities associated with urinary incontinence and pelvic organ prolapse, the majority of women do not seek medical attention and continue to suffer in silence [16]. A large prospective review of 64,650 women aged 36–55 in the Nurses’ Health Study showed that the incidence of urinary incontinence is high, but only 38% patients reported their symptoms to their physician, and only 13.9% received treatment [17]. Similarly, another study showed that 15% of patient suffering from stress urinary incontinence (SUI) sought help, but only one third of them received the recommended treatments [11]. 

There are limited studies on the prevalence of pelvic floor disorder during pregnancy. Mørkved S and Bø K studied 144 women during pregnancy and postpartum by a structured interview and clinical assessment and found that the prevalence of urinary incontinence was up to 42% The prevalence remained at 38% after 8 weeks postpartum [7]. In our study, the most prevalent item group was UDI-6, in which prevalence within the group for urinary incontinence was as high as 62.7%. This was followed by POPDI-6 (41.7%) and CRADI-8 (37.8%). In contrast to a study by Yohay et al. in 2016, the prevalence of UI, POP and FI was 41.8%, 29.0% and 24.2% respectively, which was lower than in our study. Meanwhile, Lin YH et al. studied a total of 866 women delivering their new-borns at a tertiary centre, reporting a prevalence of SUI at 51.5%, while 12.5% continued to have the same problem 12 months postpartum [18]. 

In another study by Chan et al., over 442 women demonstrated that the prevalence of UI and prolapse symptoms significantly increased with advancing gestation. In their study, the prevalence of stress urinary incontinence in the third trimester was 37.8% as compared to 9.1% in first trimester. Similarly, for urge incontinence, the prevalence was 4.9% in the first trimester and increased up to 14.3% in the third trimester [19]. In the same study, Chan et al. stated that as pregnancy advanced, the bladder neck, cervix and the ano-rectal junction descent increased significantly, and the hiatal area enlarged significantly [19]. Unfortunately, our study did not subanalyse the prevalence according to different semesters.

Primigravida poses a significant risk for urinary incontinence. A local study at Klang Valley among 306 women in their third trimester showed that the prevalence of UI was 34.3%. SUI was the most common (64.8%), followed by mixed incontinence (24.8%) and then urge incontinence (6.7%) [20]. Previous studies have shown that multiple vaginal deliveries could cause denervation of the pelvic floor and direct injury to the muscles and connective tissues, which might lead to the development of pelvic floor disorder symptoms [21]. All the differences, including different population, study design and subgroup analysis, might explain why the prevalence may differ between all the studies. 

According to Van Brummen et al., in a prospective study of over 344 women, 24.2% reported to have moderate to greatly bothersome urinary frequency symptoms at 36 weeks of gestation [5]. In their study, 35.2% participants reported moderate to greatly bothersome urinary frequency. During pregnancy, lower urinary tract symptoms became more bothersome with increasing gestational age and higher parity [5]. The symptoms of frequency and urgency during pregnancy can arise because of hypersensitivity of the bladder stretching mechanism, but they can also occur when the bladder contracts inappropriately due to detrusor overactivity [5]. Other possible mechanisms include the combination of pressure effects of the gravid uterus, altered urine production, increased glomerular filtration rate, temporary changes in the urethrovesical angle and a change in bladder capacity [5]. 

### 4.2. Knowledge and Awareness of Pelvic Floor Disorders among Pregnant Women

Overall, the median knowledge scores for PIKQ—Urinary Incontinence and PIKQ—Prolapse were 6.0 and 7.0, respectively, while the median total PIKQ score was 12.0. There were 341 (80.4%) pregnant women that had a low level of knowledge for UI and 191 (45.0%) that had a low level of knowledge for POP. 

In a recent study by Mckay et al. in 2018, over 399 diverse pregnant and postpartum women demonstrated a lack of knowledge among 74.2% of their participants regarding UI and POP [22]. In the same study, a lack of UI knowledge proficiency was most common among those of primiparity, Hispanic women, and a lower education level. A lack of prolapse knowledge proficiency was also associated with lower education level and with those women who had not seen a urologist/urogynaecologist before [22].

In another study by O’Neill AT et al., the average knowledge score was low, at 45%, for all domains. Knowledge scores were positively associated with education level and the use of books as a source on pregnancy and delivery. There were only 35% cited antenatal classes as a source [23]. Similarly, in our study, we demonstrated that pregnant women with a tertiary level of education were more likely to have better proficiency in PIKQ—UI (80.7% vs. 48.7%, *p* = 0.000) and PIKQ—POP (65.2% vs. 42.4%, *p* = 0.000). Those who received antenatal specialist care also appeared to have better knowledge in both PIKQ—UI (56.6% vs. 25.2%, *p* = 0.000) and PIKQ—POP (41.6% vs. 18.8%, *p* = 0.000).

Consistent with the study by Dunivan et al., pregnant women with a higher education were found to be associated with a proficiency in knowledge of UI and POP [24]. Dunivan et al. also noted that the greatest barrier to care seeking was cost and inconvenience [24]. Most pregnant women who received antenatal specialist care had better knowledge regarding the pelvic floor and UI. 

Several studies showed that pelvic floor muscle damage is related to mode of delivery, foetal weight and parity [25]. One reason is due to the increase in abdominal pressure, which leads to an increase in vesical pressure and urethral mobility. During delivery, the entire pelvic floor muscle undergoes huge deformation, in which the levator ani muscle and the pubococcygeus become seriously stretched [25]. Unfortunately, we did not assess the risk factors associated with pelvic floor muscle weakness in this study. 

The strengths of this study include the larger sample size, used of validated assessment tools including PFDI-20 and PIKQ questionnaires, and high response rate, among others. Our data suggest an alarming need for health care providers to bridge the knowledge gaps in PFD. Pregnancy is a great opportunity for educational intervention, as women are motivated to change their behaviour through increased risk perception and emotional references [26]. One third of women stated that they are willing to participate in a pelvic floor training program if they are given such an opportunity [27]. A previous study by Jaafar et al. showed that the use of newly developed user-centred-designed Kegel Exercise Pregnancy Training (KEPT) application is feasible and can help to constantly motivate women to adhere to pelvic floor muscle training [28]. This study hopefully serves as a basic platform for a future educational program and pelvic floor muscle training to fulfil the needs to increase knowledge and awareness about pelvic floor disorder [29]. 

However, there were a few limitations in this study. We did not assess the PFDI-20 after delivery due to time constraints. It would have been great if we could compare the PFDI-20 during pregnancy and at least 3 months postpartum. Second, the study population was based on single tertiary centre; hence, the results might not represent the entire population in Malaysia. 

## 5. Conclusions

Pelvic floor disorder during pregnancy was common. The knowledge about pelvic floor disorder was lacking among pregnant women in this study. Having tertiary education and receiving antenatal specialist care were associated with better knowledge proficiency. This study hopefully serves as a basic platform for a future educational program to bridge the knowledge gaps in PFD among women.

## Figures and Tables

**Table 1 ijerph-19-08314-t001:** Socio-demographic data of patients.

	N = 424
Age, years	31.5 (29.0, 35.0)
Ethnicity, n (%)	
• Malay	331 (78.1)
• Chinese	67 (15.8)
• Indian	9 (2.1)
• Other	17 (4.0)
Married, n (%)	417 (98.3)
Primiparity	141 (33.3)
Gestational age, weeks	36.1 (32.0, 38.2)
Body Mass Index (kg/m^2^)	27.0 (24.0, 314)
Maternal educational level, n (%)	
• Primary	1 (0.2)
• Secondary	111 (26.2)
• College	79 (18.6)
• Tertiary	233 (55.0)
Antenatal class, n (%)	85 (20.0)
Level of antenatal care, n (%)	
• Local clinic	245 (57.8)
• O&G MO/trainee	46 (10.8)
• Specialist care	133 (31.4)
Previous gynaecological surgery	14 (8.3)

Data are expressed in median (quartile), unless specified.

**Table 2 ijerph-19-08314-t002:** Results of the PFDI-20 questionnaire divided, according to symptoms, into 6 items evaluating POPDI, 8 items evaluating CRADI, and 6 items evaluating UDI.

PFDI Item		Prevalence within the Group	Frequency
1	POPDI	41.8%	232 (54.7)
2			205 (48.3)
3			147 (34.7)
4			147 (34.7)
5			202 (37.6)
6			112 (26.4)
7	CRADI	37.8%	167 (39.4)
8			179 (42.2)
9			127 (30.0)
10			135 (31.8)
11			148 (34.9)
12			141 (33.3)
13			208 (49.1)
14			138 (32.5)
15	UDI	62.7%	266 (62.7)
16			199 (46.9)
17			231 (54.4)
18			188 (44.3)
19			146 (34.4)
20			177 (41.7)
21			152 (35.8)
22			186 (43.9)

**Table 3 ijerph-19-08314-t003:** Questionnaire regarding diagnosis and severity of PFD (PFDI-20).

Bil	Questions	NO	Yes. If Yes, How Much Does This Bother You?			
			Not at All	Somewhat	Moderately	Quite a Bit
POPDI-6						
1.	Do you usually experience PRESSURE in the lower abdomen?	192 (45.3)	57 (13.4)	56 (13.2)	55 (13.0)	64 (15.1)
2.	Do you usually experience HEAVINESS or DULLNESS in the pelvic are?	219 (51.7)	63 (14.9)	35 (8.3)	55 (13.0)	52 (12.3)
3.	Do you usually have a bulge or something falling out that you can see or feel in the vaginal area?	277 (65.3)	87 (20.5)	18 (4.2)	26 (6.1)	16 (3.8)
4.	Do you usually have to push on the vagina or around the rectum to have or complete a bowel movement?	277 (65.3)	78 (18.4)	26 (6.1)	21 (5.0)	22 (5.2)
5.	Do you usually experience a feeling of incomplete bladder emptying?	222 (52.4)	59 (13.9)	50 (11.8)	45 (10.6)	48 (11.3)
6.	Do you ever have to push up on a bulge in the vaginal area with your fingers to start or complete urination?	312 (73.6)	78 (18.4)	15 (3.5)	13 (3.1)	6 (1.4)
CRADI-8						
7.	Do you feel you need to strain too hard to have a bowel movement?	257 (60.6)	72 (17.0)	34 (8.0)	44 (10.4)	17 (4.0)
8.	Do you feel you have not completely emptied your bowels at the end of a bowel movement?	245 (57.8)	68 (16.0)	44 (10.4)	38 (9.0)	29 (6.8)
9.	Do you usually lose stool beyond your control if your stool is well formed?	297 (70.0)	79 (18.6)	20 (4.7)	17 (4.0)	11 (2.6)
10.	Do you usually lose stool beyond your control if your stool is loose or liquid?	289 (68.2)	74 (17.5)	17 (4.0)	23 (5.4)	21 (5.0)
11.	Do you usually lose gas from the rectum beyond your control?	276 (65.1)	67 (15.8)	21 (5.0)	33 (7.8)	27 (6.4)
12.	Do you usually have pain when you pass your stool?	283 (66.7)	67 (15.8)	23 (5.4)	28 (6.6)	23 (5.4)
13.	Do you experience a strong sense of urgency and have to rush to the bathroom to have a bowel movement?	216 (50.9)	72 (17.0)	47 (11.1)	47 (11.1)	42 (9.9)
14.	Does a part of your bowel ever pass through the rectum and bulge outside during or after a bowel movement?	286 (67.5)	82 (19.3)	23 (5.4)	17 (4.0)	16 (3.8)
UDI-6						
15.	Do you usually experience frequent urination?	158 (37.3)	54 (12.7)	63 (14.9)	80 (18.9)	69 (16.3)
16.	Do you usually experience urinary leakage associated with a feeling of urgency, that is, a strong sensation of needing to go to the bathroom?	225 (53.1)	64 (15.1)	47 (11.1)	42 (9.9)	46 (10.8)
17.	Do you usually experience urinary leakage related to coughing, sneezing, or laughing?	193 (45.5)	57 (13.4)	62 (14.6)	54 (12.7)	58 (13.7)
18.	Do you usually experience small amounts of urinary leakage, that is, drops?	236 (55.7)	52 (12.3)	54 (12.7)	46 (10.8)	36 (8.5)
19.	Do you usually experience difficulty emptying your bladder?	278 (65.6)	72 (17.0)	28 (6.6)	25 (5.9)	21 (5.0)
20.	Do you usually experience PAIN or DISCOMFORT in the lower abdomen or genital region?	247 (58.3)	71 (16.7)	38 (9.0)	39 (9.2)	29 (6.8)
21.	Do you have pain with sexual intercourse?	272 (64.2)	64 (15.1)	36 (8.5)	32 (7.5)	20 (4.7)
22.	Have you noticed that your interest in sex has decreased?	238 (56.1)	71 (16.7)	37 (8.7)	48 (11.3)	30 (7.1)

**Table 4 ijerph-19-08314-t004:** Prolapse and Incontinence Knowledge Questionnaire—Urinary Incontinence (PIKQ—UI).

Bil	Questions	Correctly Answered	Wrongly Answered	Unsure
1.	Urinary incontinence (loss of urine or leaky bladder) is more common in younger women than in older women. (PATHOGENESIS)	218 (51.4)	78 (18.4)	128 (30.2)
2.	Women are more likely than men to leak urine. (PATHOGENESIS)	271 (63.9)	39 (9.2)	114 (26.9)
3.	Other than pads and diapers, not much can be done to treat the leakage of urine. (TREATMENT)	136 (32.1)	146 (34.4)	142 (33.5)
4.	It is NOT important to diagnose the type of urinary leakage before trying to treat it. (DIAGNOSIS)	271 (63.9)	54 (12.7)	99 (23.3)
5.	Many things can cause urinary leakage. (PATHOGENESIS)	314 (74.1)	22 (5.2)	88 (20.8)
6.	Certain exercises can be performed to help control urinary leakage. (TREATMENT)	314 (74.1)	18 (4.2)	92 (21.7)
7.	Some medications may cause urinary leakage. (PATHOGENESIS)	158 (37.3)	31 (7.3)	235 (55.4)
8.	Once people start to leak urine, they are never able to control their urine again. (TREATMENT)	228 (53.8)	57 (13.4)	139 (32.8)
9.	Doctors can perform special types of bladder testing to diagnose urinary leakage. (DIAGNOSIS)	249 (58.7)	12 (2.8)	163 (38.4)
10.	Surgery is the only treatment for urinary leakage. (TREATMENT)	161 (38.0)	45 (10.6)	218 (51.4)
11.	Giving birth many times may lead to urinary leakage. (PATHOGENESIS)	169 (39.9)	73 (17.2)	182 (42.9)
12.	Most people who leak urine can be cured or can improve with some kind of treatment. (TREATMENT)	303 (71.5)	7 (1.7)	114 (26.9)
Median number of people in each group		232	48	142
Median Score PIKQ—Urinary Incontinence		7.0 (5.0, 9.0)		

**Table 5 ijerph-19-08314-t005:** Prolapse and Incontinence Knowledge Questionnaires—Pelvic Organ Prolapse (PIKQ—POP).

Bil	Questions	Correctly Answered	Wrongly Answered	Unsure
1.	Pelvic organ prolapse (bulging of the vagina, uterus, bladder, or rectum) is more common in younger women than in older women. (PATHOGENESIS)	181 (42.7)	68 (16.0)	175 (41.3)
2.	Giving birth many times may lead to pelvic organ prolapse. (PATHOGENESIS)	208 (49.1)	50 (11.8)	166 (39.2)
3.	Pelvic organ prolapse can happen at any age. (PATHOGENESIS)	287 (67.7)	21 (5.0)	116 (27.4)
4.	Certain exercises can help to stop pelvic organ prolapse from getting worse. (TREATMENT)	293 (69.1)	12 (2.8)	119 (28.1)
5.	Symptoms of pelvic organ prolapse may include pelvic heaviness and/or pressure. (DIAGNOSIS)	256 (60.4)	10 (2.4)	158 (37.3)
6.	A good way for a doctor to diagnose pelvic organ prolapse is by examining the patient. (DIAGNOSIS)	330 (77.8)	10 (2.4)	84 (19.8)
7.	Once a patient has pelvic organ prolapse, not much can be done to help her. (TREATMENT)	221 (52.1)	38 (9.0)	165 (38.9)
8.	Heavy lifting on a daily basis can lead to pelvic organ prolapse. (PATHOGENESIS)	260 (61.3)	16 (3.8)	148 (34.9)
9.	Surgery is one type of treatment for pelvic organ prolapse. (TREATMENT)	166 (39.2)	44 (10.4)	214 (50.5)
10.	Doctors can run a blood test to diagnose pelvic organ prolapse. (DIAGNOSIS)	79 (18.6)	98 (23.1)	247 (58.3)
11.	A rubber ring called a pessary can be used to treat symptoms of pelvic organ prolapse. (TREATMENT)	113 (26.7)	15 (3.5)	296 (69.8)
12.	People who are obese are less likely to get pelvic organ prolapse. (PATHOGENESIS)	96 (22.6)	71 (16.7)	257 (60.6)
Median number of people in each group		207	37	178
Median Score PIKQ—Pelvic Organ Prolapse		6.0 (4.0, 8.0)		
Total overall score PIKQ		12.0 (8.0, 17.0)		

**Table 6 ijerph-19-08314-t006:** Level of knowledge among participants with regard to PIKQ—UI and PIKQ—POP.

		PIKQ—UI	PIKQ—POP
Level of knowledge, n (%)	Proficiency	83 (19.6)	233 (55.0)
	Low	341 (80.4)	191 (45.0)

**Table 7 ijerph-19-08314-t007:** Maternal characteristics and level of knowledge (PIKQ—UI and PIKQ—POP).

Characteristics	PIKQ—UI			PIKQ—POP		
	Proficiency (n = 83)	Low (n = 341)	Odd Ratios (CI) *p* Value	Proficiency (n = 233)	Low (n = 191)	Odd Ratios (CI) *p* Value
Age < 35 years old	61 (73.5)	270 (79.2)	0.729 (0.419–1.268) 0.300	175 (75.1)	156 (81.7)	0.667 (0.422–1.085) 0.125
Married	80 (96.4)	337 (98.8)	0.317 (0.069–1.442) 0.139	228 (97.9)	189 (98.9)	0.483 (0.093–2.515) 0.465
Primiparity	27 (32.5)	114 (33.4)	0.960 (0.576–1.601) 1.000	80 (34.3)	61 (31.9)	1.114 (0.742–1.674) 0.606
Obesity	34 (41.0)	92 (27.0)	1.863 (1.129–3.074) 0.016	78 (33.5)	48 (25.1)	1.455 (0.949–2.230) 0.087
Tertiary level education	67 (80.7)	166 (48.7)	4.415 (2.459–7.926) 0.000	152 (65.2)	81 (42.4)	2.548 (1.719–3.778) 0.000
Antenatal specialist care	47 (56.6)	86 (25.2)	3.871 (2.352–6.371) 0.000	97 (41.6)	36 (18.8)	3.071 (1.965–4.799) 0.000
Antenatal class	19 (22.9)	66 (19.3)	1.237 (0.694–2.205) 0.449	52 (22.3)	33 (17.3)	(1.376 (0.846–2.235) 0.233

Data analysis performed with Chi-square test.

## Data Availability

The data presented in this study are available on request from the corresponding author. The data are not publicly available due to ownership belonging to the institution where the study was conducted.

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
