# Peer review of "Prevalence, Knowledge and Awareness of Pelvic Floor Disorder among Pregnant Women in a Tertiary Centre, Malaysia"

_ijerph, 2022, doi:10.3390/ijerph19148314_

Round 1
Reviewer 1 Report
Abstract
1. Can authors structure the abstract? Introduction, methods, findings
2. It should be made clear in the abstract that pregnant women were recruited into the study
3. Did authors test the association between parity and symptoms of pelvic floor disorder e.g. urinary incontinence
Data analysis
Authors should provide more information on data analysis: e.g which descriptive variables were analysed using Student’s T-test and Pearson’s Chi-square test? Similarly, which continuous variables were analysed using mean and sd? What about test of association between variables? Any consideration for confounding factors?
Author Response
Please see the atttachment.

Reviewer 2 Report
The work touches on an important aspect of women's comfort. Most of the complaints described are related to postpartum situations, while constipation is a more characteristic complaint during pregnancy [Boyle et al., 2008]. In general, one encounters a lower incidence of UI, POP and FI in pregnant women, which the authors include in their discussion. Undoubtedly, to some extent, proper education of pregnant women and training of Kegel muscles before delivery would prevent the occurrence of the aforementioned complaints after delivery, as highlighted in the conclusions. In this respect, the article fulfills its role.
The paper needs some minor revisions before publication:
Pg. 2, row 68: the abbreviation SUI used for the first time should be explained;
Pg. 3, row 126: "old" redundant;
Pg. 3, row 126-134: the text is redundant, "the characteristics of the female participants in the study are given in Table I" would suffice;
Pg. 4. row 141: rename the table as above;
Pg. 11, row 245: "but" or "and" - one of them is redundant.
